# Epstein-Barr Virus miR-BART1-3p Regulates the miR-17-92 Cluster by Targeting E2F3

**DOI:** 10.3390/ijms222010936

**Published:** 2021-10-10

**Authors:** Myung Chan Park, Hyoji Kim, Hoyun Choi, Mee Soo Chang, Suk Kyeong Lee

**Affiliations:** 1Department of Medical Life Sciences, Department of Biomedicine & Health Sciences, College of Medicine, The Catholic University of Korea, Seoul 06591, Korea; pmcreg@catholic.ac.kr (M.C.P.); kims890@hanmail.net (H.K.); summons@empas.com (H.C.); 2Department of Pathology, Seoul National University Boramae Hospital, Seoul National University College of Medicine, Seoul 06591, Korea; meesooch@snu.ac.kr

**Keywords:** Epstein-Barr virus, gastric carcinoma, cell cycle, BART miRNAs, MIR17HG, miR-17-92 cluster, E2F3

## Abstract

Epstein-Barr virus (EBV) is associated with several tumors and generates BamHI A rightward transcript (BART) microRNAs (miRNAs) from BART transcript introns. These BART miRNAs are expressed at higher levels in EBV-associated epithelial malignancies than in EBV-infected B lymphomas. To test the effects of EBV miRNA on the cell cycle and cell growth, we transfected miR-BART1-3p, a highly expressed EBV-associated miRNA, into gastric carcinoma cells. We found that miR-BART1-3p induced G0/G1 arrest and suppressed cell growth in gastric carcinoma cells. As our microarray analyses showed that E2F3, a cell cycle regulator, was inhibited by EBV infection, we hypothesized that miR-BART1-3p regulates E2F3. Luciferase assays revealed that miR-BART1-3p directly targeted the 3′-UTR of E2F3 mRNA. Both E2F3 mRNA and encoded protein levels were reduced following miR-BART1-3p transfection. In contrast, E2F3 expression in AGS-EBV cells transfected with a miR-BART1-3p inhibitor was enhanced. As E2F3 has been shown to regulate the expression of highly conserved miR-17-92 clusters in vertebrates, we examined whether this expression is affected by miR-BART1-3p, which can downregulate E2F3. The expression of E2F3, miR-17-92a-1 cluster host gene (MIR17HG), and miR-17-92 cluster miRNAs was significantly reduced in EBV-associated gastric carcinoma (EBVaGC) patients compared with EBV-negative gastric carcinoma (EBVnGC) patients. Further, miR-BART1-3p as well as the siRNA specific to E2F3 inhibited the expression of the miR-17-92 cluster, while inhibition of miR-BART1-3p enhanced the expression of the miR-17-92 cluster in cultured GC cells. Our results suggest a possible role of miR-BART1-3p in cell cycle regulation and in regulation of the miR-17-92 cluster through E2F3 suppression.

## 1. Introduction

The Cancer Genome Atlas (TCGA) network carried out molecular biological analyses of clinical samples from 295 global early gastric cancer patients [1]. The gastric cancer cases were classified into four types based on the molecular characteristics of chromosomal instability, microsatellite instability, presence of Epstein-Barr virus (EBV), and genomic stability [1]. Of these, EBV-associated gastric carcinoma (EBVaGC) accounted for 9% of total stomach cancer cases [2]. In EBVaGC, EBV establishes latency type I infection and expresses a limited number of viral genes: EBV-encoded small RNAs (EBERs), BamHI A rightward transcripts (BARTs), Epstein-Barr nuclear antigen 1 (EBNA1), and latent membrane protein 2A (LMP2A). The well-known EBV oncogenes EBNA2, EBNA3s, and LMP1 are not expressed in EBVaGC [3,4].

EBV expresses its own viral microRNAs (miRNAs), and 25 pre-miRNAs have been detected in EBV-infected cells [5,6,7]. Four mature BamHI fragment H rightward open reading frame 1 (BHRF1) miRNAs generated from BHRF1 transcripts were reported to be expressed during EBV latency type III, which is observed in lymphoblastoid cell lines. In contrast, 44 BART miRNAs generated from BART transcripts are expressed in all EBV-infected cells, regardless of latent or lytic status. The expression of BART miRNA is greater in EBV-infected epithelial cells than in EBV-infected B cells [6,8]. Thus, the role of BART miRNAs in EBV-associated carcinomas such as nasopharyngeal carcinoma (NPC) and GC has attracted a great deal of attention. Despite a considerable effort to determine the function of BART miRNAs, their role in tumorigenesis and maintenance of EBVaGC remains largely unclear. The EBV miRNA miR-BART1-3p is one of the most highly expressed miRNAs in EBV-associated carcinomas such as GC and NPC [9,10]. Recently, we showed that transfection of a miR-BART1-3p mimic into gastric carcinoma cells caused the cells to accumulate at the G0/G1 phase and suppressed cell apoptosis by suppressing DAB2 [11].

The E2F family of transcription factors is engaged in fine regulation of the cell cycle, and its dysregulation is observed in a variety of human tumors. E2F1, E2F2, and E2F3 of the E2F family function as trans-activators of E2F target genes and are involved in the G1/S transition [12]. The expression of E2F3 has been shown to be regulated by cellular miRNAs [13,14]. For example, miR-145-5p was reported to directly downregulate the expression of E2F3 by seed-matching the 3′-untranslated region (UTR) of E2F3 in osteosarcoma cells, and cell growth was inhibited and G0/G1 arrest occurred when E2F3 was suppressed by miR-145-5p. Furthermore, most of the effects exhibited by miR-145-5p were absent when the expression of E2F3 was recovered using an E2F3 expression vector [14].

E2F3 interacts extensively with the miR-17-92 cluster. Endogenous E2F3 activates transcription of the miR-17-92 cluster by binding directly to its promoter, and miR-20a, a member of the mir-17-92 cluster, targets the 3′-UTR of E2F3 [15,16,17]. The miR-17-92 cluster, which is involved in multiple developmental and pathogenic processes, is highly expressed in lymphoproliferative disorders [18,19]. The function of the miR-17-92 cluster in relation to proliferation and differentiation has been extensively studied in B cells, T cells, NK cells, macrophages, and dendritic cells [20]. However, the role and regulatory mechanism of the miR-17-92 cluster in EBVaGC are largely unknown.

In this study, we investigated whether EBV miR-BART1-3p regulates the cell cycle by suppressing E2F3 expression in EBVaGC. In addition, the effects of miR-BART1-3p and E2F3 on the expression of miR-17-92 cluster were assessed.

## 2. Results

### 2.1. miR-BART1-3p Induces G0/G1 Arrest and Hinders the S-Phase Transition

MTT assays were carried out in AGS and AGS-EBV cells transfected with a miR-BART1-3p mimic or inhibitor of miR-BART1-3p (miR-BART1-3p(i)). Proliferation of AGS cells was significantly reduced by miR-BART1-3p transfection relative to that of the scrambled control transfection (Figure 1A). In addition, cell growth of AGS-EBV cells was suppressed following transfection with miR-BART1-3p(i) relative to the control inhibitor (Figure 1B). To test whether miR-BART1-3p regulates the cell cycle, we accessed the cell-cycle distribution of transfected cells after propidium iodide (PI) staining. Compared with cells transfected with the scrambled control, miR-BART1-3p mimic transfection increased the proportion of cells in the G0/G1 phase and decreased the proportions in the S and sub-G1 phases (Figure 1C,D). In contrast, AGS-EBV cells transfected with miR-BART1-3p(i) had a reduced G0/G1-phase population, while the S and sub-G1 phase populations were increased compared with cells transfected with the control inhibitor (Figure 1E,F). The proportion of G2/M-phase cells was not influenced by transfection with the miR-BART1-3p mimic or miR-BART1-3p(i) (Figure 1C,E).

### 2.2. E2F3 Expression Is Suppressed in EBVaGC- and EBV-Infected Cells

The G0/G1 transition is regulated by E2F1, E2F2, and E2F3, belonging to the E2F family [12]. As transfection of miR-BART1-3p induced cell-cycle arrest at the G0/G1 phase in the AGS cells, we investigated which E2F family member is involved in the cell-cycle arrest following miR-BART1-3p transfection.

Expression of E2F1, E2F2, and E2F3 was compared in tissue samples from EBVaGC and EBV-negative gastric carcinoma (EBVnGC) patients using the Gene Expression Omnibus (GEO) database with the accession number GSE51575. The GEO data showed lower E2F1 and E2F3 expression in EBVaGC than in EBVnGC tissue (Figure 2A). However, E2F2 expression was not significantly different between EBVaGC and EBVnGC tissue samples (Figure 2A). We performed qRT-PCR to compare the mRNA expression of E2F1, E2F2, and E2F3 in AGS and AGS-EBV cells. The results showed that E2F3 had lower mRNA expression in AGS-EBV than in AGS cells, while E2F1 and E2F2 mRNA expression did not differ between the two cell types (Figure 2B). E2F3 protein level was lower in AGS-EBV than in AGS cells (Figure 2C).

### 2.3. miR-BART1-3p Suppresses E2F3 Expression

As only E2F3 expression was reduced by EBV infection in both the GEO analyses and the cell lines, we assessed whether the reduction in E2F3 expression was related to miR-BART1-3p. First, the effect of miR-BART1-3p on E2F3 expression was examined. Expression of both E2F3 mRNA and protein was reduced in AGS cells following transfection with the miR-BART1-3p mimic relative to the scrambled-control-transfected cells (Figure 3A,B). In contrast, E2F3 mRNA and protein were increased in AGS-EBV cells transfected with miR-BART1-3p(i) relative to cells transfected with the control inhibitor (Figure 3C,D).

For comparison, the effect of the siRNA specific for E2F3 (siE2F3) on cell-cycle distribution was assessed following propidium iodide (PI) staining. Compared with transfection with the scrambled control, siE2F3 transfection increased the proportion of cells in the G0/G1 phase and reduced the proportions of cells in the S and sub-G1 phases. The proportion of the G2/M-phase population was reduced by siE2F3 (Figure 3E,F).

### 2.4. miR-BART1-3p Directly Targets the 3′-UTR of E2F3

To determine whether E2F3 is a direct target of miR-BART1-3p, luciferase reporter assays were carried out using HEK293T cells co-transfected with 3′-UTR luciferase reporter constructs (psiC-E2F3) and the miR-BART1-3p mimic or mutated miR-BART1-3p with nucleotides 1 to 3 substituted (miR-BART1-3pm) (Figure 4 top panel). Figure 4A indicates that miR-BART1-3p and the E2F3 mRNA have a non-canonical seed match. The luciferase activity of psiC-E2F3 was significantly reduced by the miR-BART1-3p mimic relative to the psiCHECK control in transfected HEK293 cells (Figure 4B). The luciferase reporter assay was repeated in AGS cells, and similar results were obtained (Figure 4C). Transfection with miR-BART1-3pm did not affect luciferase activity in either AGS or HEK239T cells (Figure 4B,C). Therefore, the E2F3 gene was selected as a miR-BART1-3p target and further analyzed.

To confirm specific binding of miR-BART1-3p to E2F3, the seed match sequences in psiC-E2F3 were mutated to produce psiC-E2F3m by site-directed mutagenesis (Figure 4A bottom panel). The psiC-E2F3 or psiC-E2F3m was co-transfected with miR-BART1-3p or miR-BART1-3pm into AGS cells for luciferase reporter assays. As expected, miR-BART1-3p suppressed luciferase activity of psiC-E2F3 in AGS cells (Figure 4D). In contrast, luciferase activity was not altered by miR-BART1-3p when AGS cells were transfected with the mutant psiC-E2F3m reporter construct. There was no alteration in luciferase activity owing to miR-BART1-3pm regardless of the co-transfected reporter construct (Figure 4D). We also performed luciferase assays in AGS-EBV cells co-transfected with each of the luciferase reporter constructs and miR-BART1-3p(i) (Figure 4E). When AGS-EBV cells were co-transfected with psiC-E2F3 and the control inhibitor, there was significant reduction in luciferase activity, indicating that endogenously expressed miR-BART1-3p was sufficient to suppress the expression of the reporter construct. Furthermore, recovery of luciferase activity was evident when the cells were co-transfected with psiC-E2F3 and miR-BART1-3p(i) (Figure 4E). When psiC-E2F3m was transfected into AGS-EBV cells, luciferase activity was unchanged and was not affected by the co-transfected inhibitor. These results support a role for miR-BART1-3p in directly targeting the 3′-UTR of E2F3.

### 2.5. miR-17-92 Cluster Expression is Suppressed in EBVaGC and EBV-Infected Cells

Previous studies have shown that E2F3 binds to the promoter of the miR-17-92 cluster and activates its transcription [16]. As E2F3 expression was suppressed in EBVaGC and AGS-EBV cells relative to EBV-negative counterparts, we examined whether reduced E2F3 expression affected the expression of the miR-17-92 cluster. The expression of miR-17-92a-1 cluster host gene (MIR17HG) in AGS-EBV cells was lower than in AGS cells (Figure 5A). Data for EBVnGC and EBVaGC patients obtained from the GEO database (GSE51575) were also analyzed. The results indicated that MIR17HG expression was lower in EBVaGC patients than in EBVnGC patients (Figure 5B). When the expression of the six individual miRNAs comprising the miR-17-92 cluster was assessed in the cell lines, four (miR-19a-3p, miR-20a-5p, miR-19b-3p, and miR-92a-3p) were slightly reduced in AGS-EBV compared with AGS cells (Figure 5C). However, the expression of miR-17-5p and miR-18a-5p did not differ significantly in these two cell lines (Figure 5C). The expression of individual miRNAs of the miR-17-92 cluster was assessed using RNA extracted from gastric carcinoma tissues, too. Five of the six miRNAs of the miR-17-92 cluster (miR-17-5p, miR-18a-5p, miR-19a-3p, miR-20a-5p, and miR-19b-3p) showed lower expression in EBVaGC than in EBVnGC (Figure 5D). However, the expression of miR-92a-3p did not differ significantly in EBVaGC and EBVnGC tissues.

### 2.6. miR-BART1-3p and siE2F3 Suppress miR-17-92 Cluster miRNA Expression

To determine if miR-BART1-3p regulates miR-17-92 cluster expression, the effects of miR-BART1-3p and the inhibitor of miR-BART1-3p on miR-17-92 cluster expression were examined. The AGS and AGS-EBV cells were transfected with the miR-BART1-3p mimic and miR-BART1-3p(i), respectively. Expression of MIR17HG as determined by qRT-PCR was lower in AGS cells following transfection with the miR-BART1-3p mimic than with the scrambled control (Figure 6A). Furthermore, transfection with miR-BART1-3p(i) upregulated MIR17HG expression in AGS-EBV cells (Figure 6C). To confirm these results, the expression of each miRNA encoded within the miR-17-92 cluster was determined by qRT-PCR following transfection of AGS cells with miR-BART1-3p or AGS-EBV cells with miR-BART1-3p(i). Similar to the results for MIR17HG, the expression of the individual miRNAs was downregulated by the miR-BART1-3p mimic in AGS cells (Figure 6B), but was upregulated by miR-BART1-3p(i) in AGS-EBV cells (Figure 6D). Next, the effect of E2F3 on the expression of the miR-17-92 cluster was analyzed. The AGS cells transfected with siE2F3 showed a reduction in MIR17HG expression (Figure 6E). Similarly, the expression of the individual miR-17-92 cluster miRNAs was suppressed to a greater degree by siE2F3 transfection than by the scrambled control transfection (Figure 6F).

### 2.7. Expression Level of miR-BART1-3p in Cells and Tumor Tissues

We compared the expression level of miR-BART1-3p in cells and tumor tissues by qRT-PCR (Appendix A). Expression levels were assessed by the delta Cq method, where Cq of U6 was utilized for normalization. In AGS-EBV, miR-BART1-3p showed Cq values of around 27–28, while U6 showed Cq values of around 13, suggesting that miR-BART1-3p was expressed about 2^14^^~15^ fold less compared with U6 in this cell line. In AGS cells not infected with EBV, we could not detect miR-BART1-3p, and the results were marked as “not detected (N.D.)” on the figures (Appendix A). When qRT-PCR was carried out using RNA from tumor tissue, miR-BART1-3p expression was undetectable in any of the EBVnGC tissues. In contrast, the expression of miR-BART1-3p was readily detectable in all of the EBVaGC samples and the level of miR-BART1-3p was 2^10^ times lower than that of U6 in the same tissue samples (Appendix A). In AGS cells transfected with miR-BART1-3p, the miR-BART1-3p level was about 20 times higher than that of AGS-EBV (Appendix A). When AGS-EBV cells were transfected with the miR-BART1-3p inhibitor, the level of miR-BART1-3p was decreased by more than 90% compared with the control transfected AGS-EBV cells (Appendix A).

## 3. Discussion

We found that the expression of E2F3 and miR-17-92 cluster family is reduced to a greater degree in EBVaGC than in EBVnGC. Our data showed that miR-BART1-3p suppressed the expression of E2F3 by targeting its 3′-UTR. In addition, the same miRNA hindered cell growth by arresting cell-cycle progression at the G0/G1 phase. The expression of the miR-17-92 cluster family was suppressed by miR-BART1-3p (Figure 6B) and by an siRNA specific to E2F3 (Figure 6F), but was enhanced by an inhibitor of miR-BART1-3p (Figure 6D).

Our results showed that E2F3, which can affect cell growth and apoptosis [13,21], was inhibited by miR-BART1-3p. When miR-BART1-3p was transfected into AGS cells, the relative expression level of miR-BART1-3p was about 20 times higher than in AGS-EBV cells, which was similar to that of EBVaGC. We previously reported that the expression levels of BART miRNAs were noticeably higher compared with the cellular miRNAs in paraffin-embedded EBVaGC tissues, suggesting that EBV BART miRNAs are either expressed at a higher level in vivo or are more stable in vivo than in the cultured cells [22]. In addition, the expression level of BART miRNAs was clearly higher in tumors established by implanting EBVaGC cells to nude mice than in the very implanted cells, suggesting that the expression levels of BART miRNAs are higher in in vivo than in monolayer cultured cells [23]. Thus, the level of miR-BART1-3p in the transiently transfected AGS cells was comparable with that in EBVaGC tissues, supporting that our experimental conditions were within the physiological range.

In order to regulate E2F family members, EBV seems to adopt different EBV genes depending on the EBV latency type. In B cells, the expression of EBNA3C inhibits cell growth by suppressing the activity and expression of E2F1. In addition, EBNA3C increases resistance to apoptosis caused by DNA damage [24]. We found the expression of E2F1 to not be different in EBVnGC and EBVaGC. In EBVaGC displaying latency type I EBV infection, BART miRNAs are highly expressed, while EBNA3C is not expressed [22]. Thus, EBV miR-BART1-3p, but not EBNA3C can play a role in suppressing E2Fs in EBVaGC. In contrast, in B cells such as LCLs showing latency type III EBV infection, EBNA3C is expressed, while BART miRNAs are expressed only at low levels [22]. In this setting, EBNA3C is used to modulate E2Fs. In our experiments, inhibition of E2F3 by miR-BART1-3p may have contributed to suppression of cell growth and the increase in the proportion of G0/G1 cells in AGS-EBV, which shows modified latency 1 EBV infection without EBNA3C expression. These findings suggest that miR-BART1-3p increases resistance to DNA-damage-induced apoptosis by affecting E2F3. Previously, we showed that miR-BART1-3p can inhibit cell apoptosis by acting on disabled homolog 2 (DAB2) in addition to E2F3 [11], suggesting that miRNAs fine tune cell function by targeting multiple genes with overlapping functions.

Most miRNA gene interactions occur in a region of six or seven contiguous conserved nucleotides (nucleotides 2–8 of the miRNA), termed the seed region [25]. However, studies have revealed that some miRNAs interact with target genes through non-canonical matches that contain bulged or mismatched nucleotides [26]. In a non-canonical seed match, the target bulges from position 6 allow hybridization toward the 3′ end of the miRNA. This bulge formation confers thermodynamic stability to the consecutive base pairs of nucleation [27]. In addition to the seed match, extra base pairs at nucleotides 12–17 enhance the miRNA-targeting effect [28]. The E2F3 3′-UTR and miR-BART1-3p can bind through nucleotides 13–22 in addition to nucleotides 1–8 with an A bulge in position 6. This additional binding might confer an enhanced interaction between the 3′-UTR of E2F3 and miR-BART1-3p.

The analyses of GEO data (GSE51575) for GC patients as well as our microarray results from EBV-positive and -negative GC cell lines (GSE135644) showed that the expression of MIR17HG, the host gene for the miR-17-92 cluster, was downregulated by EBV infection. In addition, the expression of miR-17-5p, miR-18a-5p, miR-19a-3p, miR-20a-5p, and miR-19b-3p was downregulated in EBVaGC compared with EBVnGC tissue samples. Unexpectedly, the expression of miR-92a-3p was not changed by EBV infection, unlike that of other miRNAs from the same miR-17-92 cluster. In the experiments using cell lines, the expression of miR-19a-3p, miR-20a-5p, miR-19b-3p, and miR-92a-3p was lower in AGS-EBV than in AGS cell lines. However, the expression of miR-17-5p and miR-18a-5p was not significantly different in the two cell lines. Processing of miRNA transcripts by Drosha/DGCR8 and dicer is required to produce mature miRNAs. During miRNA maturation, RNA-binding proteins increase the expression of specific miRNAs by recognizing specific loop structures and promoting processing by Drosha [29,30,31]. In addition, lncRNAs such as GAS5 modulate miRNA expression by inducing degradation of certain mature miRNAs or acting as an miRNA sponge [32,33]. Therefore, the differences in miRNA expression observed in our experiments might have been related to the proteins and lncRNAs involved in post-transcriptional regulation of miRNAs. Further research is needed to assess these possibilities.

The miR-17-92 cluster is one of the most studied miRNA clusters and plays either an oncogenic or a tumor-suppressive role depending on the tumor type [34,35,36]. Contrary to what we observed in EBVaGC, the miR-17-92 cluster family was previously reported to be highly expressed in NPC [37]. EBV exerts different latencies in EBVaGC and NPC. In EBVaGC, EBNA1, LMP2A, EBERs, and BART miRNAs are expressed during modified latency type I infection. In contrast, EBV establishes latency type II infection in NPC, in which LMP1 is expressed in addition to the genes expressed in EBVaGC. The LMP1 gene is known to affect several signaling pathways through its C-terminal-activating region 1 (CTAR1) and CTAR2 domains and downregulates miR-203 expression by regulating the JNK and NF-kB pathways in NPC [38,39]. Reduction of miR-203 has been reported to increase the expression of E2F3, consistent with miR-203 as an miRNA that can target E2F3 [13]. The activity of STAT3 is also increased by LMP1, another transactivator of the miR-17-92 cluster [40]. Thus, the differences in the expression of the miR-17-92 cluster in EBVaGC and NPC could be due to the differences in LMP1 expression in these two carcinomas.

We found that miR-BART1-3p and an siRNA specific for E2F3 suppressed the expression of MIR17HG and the miR-17-92 cluster. This finding implies that the reduction of the miR-17-92 cluster observed in EBVaGC is at least partially due to inhibition of E2F3 by miR-BART1-3p. There are few other reports showing EBV viral miRNA directly affecting cellular miRNA expression. However, as miR-BART5-3p inhibits p53 expression by targeting its 3′-UTR [41] and p53 is known to regulate the expression of several types of miRNAs [42], p53 inhibition by miR-BART5-3p could modulate the expression of various cellular miRNAs. Interestingly, E2F3 directly binds to and activates transcription of the miR-17-92 cluster [15,16], while the miRNAs encoded within the miR-17-92 cluster inhibit translation of E2F3 [43]. Thus, a feedback loop linking miR-BART1-3p, E2F3, and miR-17-92 cluster miRNAs might be functioning in EBVaGC. Further study is warranted to clarify whether and how this feedback loop affects EBV oncogenicity in EBVaGC.

## 4. Materials and Methods

### 4.1. Cell Lines

This study used the AGS EBV-negative gastric cancer cell line and AGS-EBV, an AGS cell line infected with a recombinant EBV virus from Akata cell line [43,44,45]. The cells were cultured in RPMI-1640 medium containing 10% fetal bovine serum (FBS), 100 U/mL penicillin, and 100 μg/mL streptomycin. To culture the AGS-EBV cells, 400 μg/mL of G418 (Gibco, Carlsbad, CA, USA) was added to the medium. The human embryonic kidney cell line HEK293T was cultured in Dulbecco’s modified Eagle medium (DMEM) supplemented with 10% FBS, 100 U/mL penicillin, and 100 μg/mL streptomycin. All cells were maintained in an incubator at 37 °C with 5% CO₂.

### 4.2. mRNA Extraction from Gastric Carcinoma Tissue Samples

The study protocol was reviewed and approved by the Institutional Review Board of the Seoul National University Boramae Hospital under conditions of anonymity (IRB No. 20140204/26-2014-13/022). All human tissue specimens were obtained during diagnostic and therapeutic surgery. This study was performed using stored paraffin blocks containing tissue samples after pathologic diagnosis, and all of the samples were anonymized before the study. The tumor portion was marked on microscopic examination, punched out from each formalin-fixed paraffin-embedded tissue block using a trephine apparatus, and then placed in 1.5 mL tubes. Total RNA including miRNA was extracted using the miRNeasy FFPE kit (Qiagen, Hilden, Germany) according to the manufacturer’s instructions. The tissue lysate was treated with DNase I (Roche, Basel, Switzerland) to exclude DNA contamination.

### 4.3. Transfection with miRNA Mimic, miRNA Inhibitor, and siRNA to E2F3

The miRNA mimics, siRNA, and the scrambled control were purchased from Genepharma (Shanghai, China). The sequences are as follows: siRNA to E2F3: 5′-GUACCUCUCAGAUGGUUUATT-3′ and the control siRNA 5′-UUCUCCGAACGUGUCACGUTT-3′. The locked nucleic acid (LNA) inhibitor of miR-BART1-3p (catalog no. 410974-00) and the negative-control LNA-miRNA inhibitor (catalog no. 199020-00) were purchased from Exiqon (Vedbaek, Denmark). The mirVana^TM^ miRNA inhibitor for miR-BART1-3p (catalog No 4464084) and the control inhibitor (catalog No 4464076) were purchased from Invitrogen (Carlsbad, CA, USA). The two inhibitors showed comparable effects under the conditions used in our experiments. The scrambled control and the control inhibitor were used as negative controls. The AGS and AGS-EBV cells were seeded into 100 mm diameter dishes containing 10 mL culture medium 24 hr prior to transfection. The cells were transfected with 30 nM each of miRNA mimics, LNA inhibitors, mirVana inhibitors, or siRNAs using Lipofectamine™ 2000 (Invitrogen, Carlsbad, CA, USA) according to the manufacturer’s protocol. Optimal concentration for each reagent was determined by testing a range of concentrations. Protein or RNA was extracted 48 hr after transfection. For the FACS analyses, the cells were harvested 48 hr after transfection.

### 4.4. Cell Proliferation Assay

Cell proliferation was assessed using a 3-(4,5-dimethylthiazol-2-yl)-2,5-diphenyltetrazolium bromide (MTT) assay (Cayman, Ann Arbor, MI, USA). The AGS and AGS-EBV cells (4 × 10^5^ cells/well) were seeded into 96-well plates. After the indicated periods following transfection, 20 µL of MTT solution (5 mg/mL) was added to each well. The absorbance at 570 nm was measured with a SoftMax apparatus (Molecular Devices, San Jose, CA, USA) 4 h after adding MTT solution.

### 4.5. Cell Cycle Analyses Using Propidium Iodide (PI) Staining

Cells were harvested, washed with PBS, and fixed in 70% ethanol at −20 °C for 16 h. The cells were washed twice with PBS and then resuspended in PBS containing 10 µg/mL RNase A (Invitrogen, Carlsbad, CA, USA) and 50 µg/mL PI (Sigma-Aldrich, St. Louis, MO, USA). The distribution of cells in each phase of the cell cycle was analyzed using a FACSCalibur™ apparatus (BD Biosciences, San Diego, CA, USA), as described previously [46].

### 4.6. Target Prediction

The E2F3 sequence used for miRNA target prediction was extracted from the National Center for Biotechnology Information database (NM_001949.5). To examine whether the 3′-UTR of E2F3 could be targeted by BART miRNAs, we used a publicly available RNA hybrid program (http://bibiserv.techfak.uni-bielefeld.de/rnahybrid/2020/12/28, accessed on 3 August 2021). This tool finds the minimum free energy of hybridization needed for miRNAs to bind to specific RNAs.

### 4.7. Plasmid Constructs

The 3′-UTR of E2F3 was amplified using the cDNA prepared from AGS cells to construct psiC-E2F3. The sequences of the primers used for the plasmid construct were as follows: 5′-TCTAGGCGATCGCTCGAGTTATGCTTCGTGTGAACTCT-3′ and 5′-TTATTGCGGCCAGCGGCCGCAACTCTTAAAAAAAATCATTTTATTGATCCT-3′. The amplicons were cloned into XhoI/NotI sites between the Renilla luciferase coding sequence and the poly(A) site of the psiCHECK-2 plasmid (Promega, Madison, WI, USA) using an EZ-Fusion™ cloning kit (Enzynomics, Daejeon, Korea). We substituted the nucleotides at the fourth, fifth, and seventh positions of the seed match sequence in psiC-E2F3 to produce psiC-E2F3m using the EZchange™ site-directed mutagenesis kit (Enzynomics, Daejeon, Korea). The primers used for this purpose were as follows: 5′-ACCTAAGGACTGCGGGAATAAGG-3′ and 5′-TGCAAGGAATGGACCGGAGATACTACG-3′.

### 4.8. Luciferase Reporter Assay

To investigate the direct effect of miRNAs on the expression of target genes, cells were seeded into 96-well plates at 3 × 10^3^ cells/well. After 24 hr, HEK293T or AGS cells were co-transfected with 20 ng psiC-E2F3 or psiC-E2F3m and the miR-BART1-3p or miR-BART1-3pm using Lipofectamine™ 2000 (Invitrogen, Carlsbad, CA, USA). Similarly, AGS-EBV cells were co-transfected with 20 ng psiC-E2F3 or psiC-E2F3m and miR-BART1-3p(i). Luciferase activity was measured 48 hr post-transfection using the Dual-Glo^®^ luciferase reporter assay system (Promega). For each sample, Renilla luciferase activity was normalized using the internal control firefly luciferase activity.

### 4.9. Quantitative Real-Time-Polymerase Chain Reaction (qRT-PCR)

The AGS and AGS-EBV cells were harvested, and total RNA was extracted using RNAiso Plus reagent (TakaRa, Tokyo, Japan) according to the manufacturer’s instruction. Then, cDNA was synthesized using 2 μg total RNA, oligo(dT) primers (Macrogen, Seoul, Korea), and Moloney murine leukemia virus (M-MLV) reverse transcriptase (Invitrogen, Carlsbad, CA, USA). The qRT-PCR for the indicated genes was carried out using TOPreal^TM^ Qpcr 2x Pre-MIX SYBR-Green (Enzynomics, Daejeon, Korea) with a qRT-PCR system (CFX96, BioRad, Hercules, CA, USA). The sequences of the primers used for gene amplification were as follows: E2F1, 5′-CATCAGTACCTGGCCGAGAG-3′ and 5′-AAGCGCTTGGTGGTCAGATT-3′; E2F2, 5′-AGTCAGAGGATGGGGTCCTG-3′ and 5′-GCCTACCCACTGGATGTTGT-3′; E2F3, 5′-GTATGATACGTCTCTTGGTCTGC-3′ and 5′-CAAATCCAATACCCCATCGGG-3′; MIR17HG, 5′-CAGTAAAGGTAAGGAGAGCTCAATCTG-3′ and 5′-CATACAACCACTAAGCTAAAGAATAATCTGA-3′; GAPDH, 5′-ATGGGGAAGGTGAAGGTCG-3′ and 5′-GGGGTCATTGATGGCAACAATA-3′. The PCR conditions were 95 °C for 10 min, followed by 35 cycles of 95 °C for 10 s, 60 °C for 30 s, and 72 °C for 30 s. To confirm specific amplification of the PCR product, dissociation curves were checked routinely. For this, reaction mixtures were incubated at 95 °C for 60 s and then ramped from 60 °C to 95 °C at a heating rate of 0.1 °C/s, with fluorescence measured continuously. Relative gene expression was calculated using quantification cycle (Cq) values, with GAPDH as an internal standard.

### 4.10. Western Blot Analysis

Cells were lysed in radioimmunoprecipitation assay (RIPA) buffer containing protease inhibitors (1 mM phenylmethylsulfonyl fluoride, 10 µg/mL leupeptin, 10 µg/mL pepstatin A, and 10 µg/mL aprotinin). Cell lysate equivalent to 50 μg protein was mixed with 5× loading buffer (Fermentas, Waltham, MA, USA) and heated at 95 °C for 5 min. Samples were separated by electrophoresis on 10% sodium dodecyl sulfate polyacrylamide gels, and the separated proteins were transferred to a polyvinylidene fluoride (PVDF) membrane (Millipore, Billerica, MA, USA). Membranes were blocked and probed with the following antibodies: mouse anti-E2F3 (1:500; Santa Cruz Biotechnology, SC-56665) and rabbit anti-β-actin (1:2000; Cell Signaling Technology, #4970S). Bound antibodies were detected with horseradish-peroxidase-conjugated anti-mouse (GeneTex, GTX213111-01, Irvine, CA, USA) and anti-rabbit (Cell Signaling Technology, #7074S) secondary antibodies at a dilution of 1:3000 for 1 h at room temperature. Protein bands were visualized using an enhanced chemiluminescence detection system (Amersham Bioscience, Piscataway, NJ, USA), and the membrane was exposed to X-ray film (Agfa, Mortsel, Belgium). Anti-β-actin antibody was used to confirm comparable loading between samples. The density of each protein band was quantified using Image J software (National Institutes of Health, Bethesda, MD, USA).

### 4.11. Microarray Analysis

Microarray data that we published previously were used for analysis [47]. In brief, the microarray analysis comparing AGS and AGS-EBV was conducted using the Sentrix^®^ Human-6 v2 Expression BeadChip (Illumina, San Diego, CA, USA) (deposited as GSE135644 in GEO). The raw mRNA microarray data from EBVaGC and EBVnGC patients were adopted from a published study (GSE51575 in GEO).

### 4.12. Quantitative Real-Time PCR for miRNA Analysis

The miRNA cDNA was synthesized using a Mir-X™ miRNA First-Strand synthesis kit (Clontech, Mountain View, CA, USA) according to the manufacturer’s instructions. Quantitative real-time PCR was performed using TOPreal^TM^ Qpcr 2× Pre-MIX SYBR-Green (Enzynomics, Daejeon, Korea). Specific miRNA sequences in the cDNA were quantified using miRNA-specific sequences as 5′ primers. The sequences of the forward primers were as follows: miR-17-5p, 5′-CAAAGTGCTTACAGTGCAGGTAG-3′; miR-18a-5p, 5′-TAAGGTGCATCTAGTGCAGATAG-3′; miR-19a-3p, 5′-TGTGCAAATCTATGCAAAACTGA-3′; miR-20a-5p, 5′-TAAAGTGCTTATAGTGCAGGTAG-3′; miR-19b-3p, 5′-TGTGCAAATCCATGCAAAACTGA-3′; and miR-92a-3p, 5′-TATTGCACTTGTCCCGGCCTGT-3′. All amplifications were performed in triplicate, and values were normalized using the U6 endogenous control, which was amplified with the control primers supplied in the kit.

### 4.13. Statistical Analysis

The data were analyzed using a one-way repeated-measures analysis of variance (ANOVA) or Student’s *t*-test. Curve fitting and analyses were performed using GraphPad Prism (GraphPad Software, San Diego, CA, USA), and *p*-values less than 0.05 were considered statistically significant. All results are expressed as the mean ± standard deviation (SD).

## Figures and Tables

**Figure 1 ijms-22-10936-f001:**
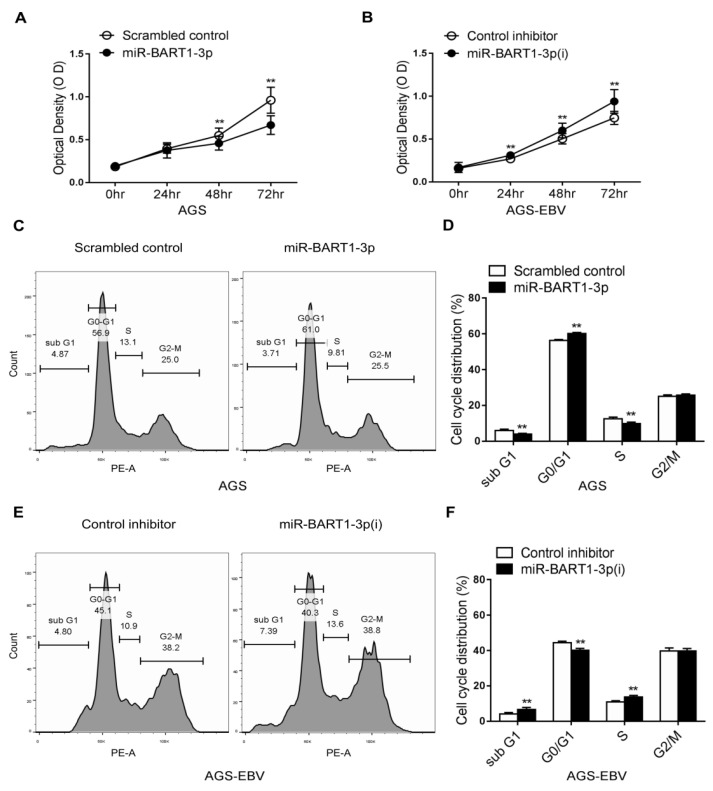
Cell cycle regulation by miR-BART1-3p. The AGS and AGS-EBV cells were transfected with either the miR-BART1-3p mimic (**A**), (**C**), and (**D**) or the miRVana-miR-BART1-3p(i) (**B**), (**E**), and (**F**). (**A**) and (**B**) At the indicated times after transfection, 20 µL of MTT solution was added to each well to assess cell proliferation. (**C**), (**D**), (**E**), and (**F**) The cells were stained with PI and subjected to FACS analysis 48 hr after transfection. Three sets of independent experiments were performed, and representative results (**C**) and (**E**) and mean +/− SD values (**D**) and (**F**) are shown. The data were analyzed using Student’s *t*-test. Error bars indicate the SD (*n* = 3). **, *p* < 0.01.

**Figure 2 ijms-22-10936-f002:**
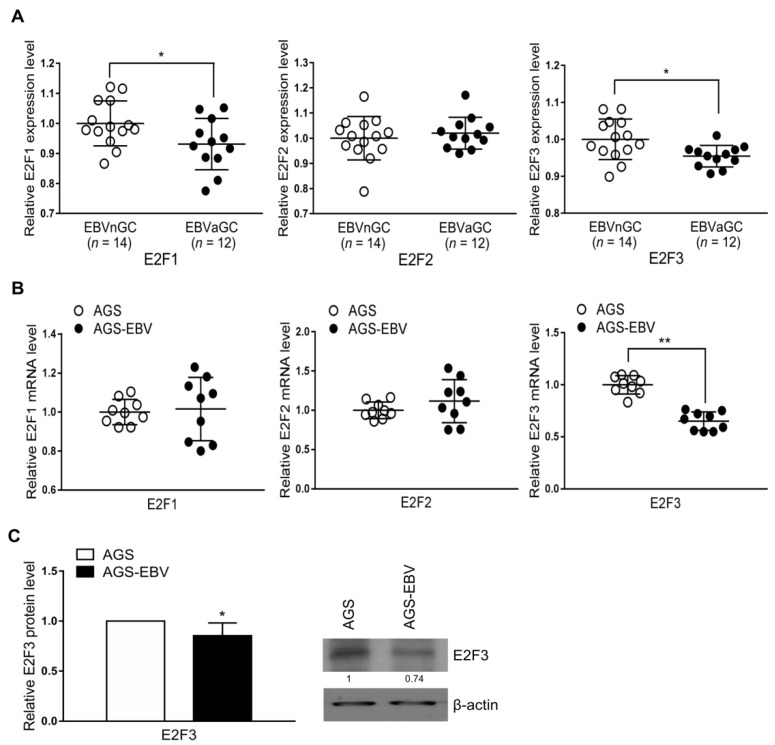
E2F1, E2F2, and E2F3 expression in EBV-negative and EBV-positive samples. (**A**) Results of GEO database analysis for E2F1, E2F2, and E2F3 expression in EBVnGC and EBVaGC patients. (**B**) Expression of E2F1, E2F2, and E2F3 mRNA in AGS and AGS-EBV cells was analyzed by qRT-PCR using a SYBR green qPCR kit. The mRNA levels were normalized to those of GAPDH. (**C**) Expression of the E2F3 protein was analyzed by Western blotting using three sets of independently prepared cells. Representative results are shown in C. The E2F3 protein level in AGS-EBV was normalized to that of β-actin and presented as a ratio compared with the values obtained from AGS cells. The data were analyzed using Student’s *t*-test. Error bars indicate the SD (*n* = 3). *, *p* < 0.05; **, *p* < 0.01.

**Figure 3 ijms-22-10936-f003:**
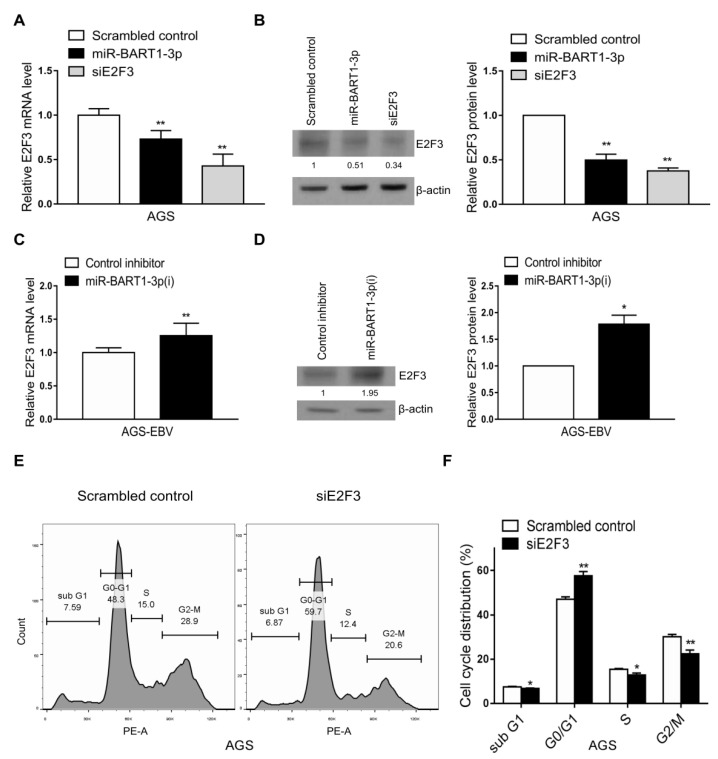
Effect of miR-BART1-3p on E2F3 expression. (**A**) and (**B**) The AGS cells were transfected with the miR-BART1-3p mimic, siE2F3, or the scrambled control. (**C**) and (**D**) The AGS-EBV cells were transfected with LNA-miR-BART1-3p(i) or the control inhibitor. The cells were harvested 48 hr after transfection for analysis. (**A**) and (**C**) Expression of E2F3 mRNA was analyzed by qRT-PCR using a SYBR green qPCR kit. (**B**) and (**D**) Expression of E2F3 protein was analyzed by Western blotting using three sets of independently transfected cells. Representative results are shown in (**B**) and (**D**). The E2F3 protein level was normalized to that of β-actin and shown as the ratio compared with the values obtained from the control transfected cells. The mean +/− SD from three independent experiments are presented. (**E**) and (**F**) The AGS cells were transfected with the siE2F3 or the scrambled control, stained with PI, and subjected to FACS analysis 48 hr after transfection. Three sets of independent experiments were performed. Representative results (**E**) and mean +/− SD values (**F**) are shown. The data were analyzed using Student’s *t*-test. *, *p* < 0.05; **, *p* < 0.01.

**Figure 4 ijms-22-10936-f004:**
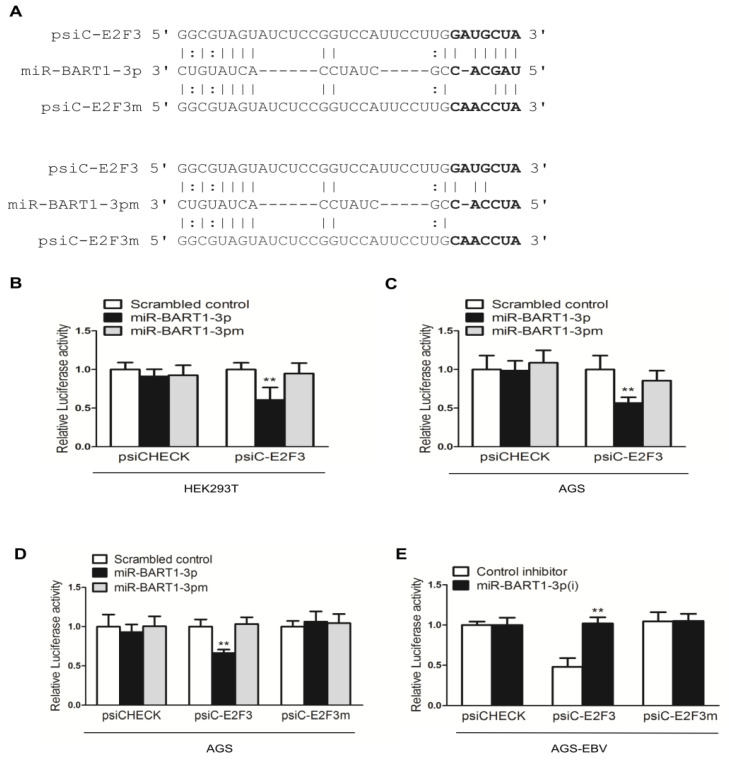
miR-BART1-3p targeting E2F3. (**A**) Seed match between miR-BART1-3p or miR-BART1-3pm and the luciferase reporter constructs containing the wild-type or the mutant 3′-UTR of E2F3. (**B**) Luciferase reporter assays. miR-BART1-3pm was used to check the specific binding between the miRNA mimics and the reporter constructs. The HEK293T cells were co-transfected with the miR-BART1-3p mimic or the miR-BART1-3pm and each of the luciferase reporter constructs containing the 3′-UTR of E2F3. The scrambled control and the empty reporter vector, psiCHECK, were included as controls. Luciferase activity was measured 48 hr after transfection and normalized using firefly activity. Error bars indicate the SD (*n* = 3). (**C**) The AGS cells were co-transfected with 30 nM miR-BART1-3p mimic or the miR-BART1-3pm, and the appropriate luciferase reporter construct. (**D**) The AGS cells were co-transfected with 30 nM miR-BART1-3p mimic or miR-BART1-3pm and 20 ng appropriate luciferase reporter construct containing the wild-type or the mutant 3′-UTR of E2F3. (**E**) The AGS-EBV cells were co-transfected with 30 nM LNA-miR-BART1-3p(i) and 20 ng appropriate luciferase reporter construct containing the wild-type or the mutant 3′-UTR of E2F3. Luciferase activity was measured 48 hr after transfection, normalized using firefly luciferase activity, and expressed as a ratio compared with that of the scrambled control-transfected cells. The data were analyzed using ANOVA. Error bars indicate the SD (*n* = 3). **, *p* < 0.01.

**Figure 5 ijms-22-10936-f005:**
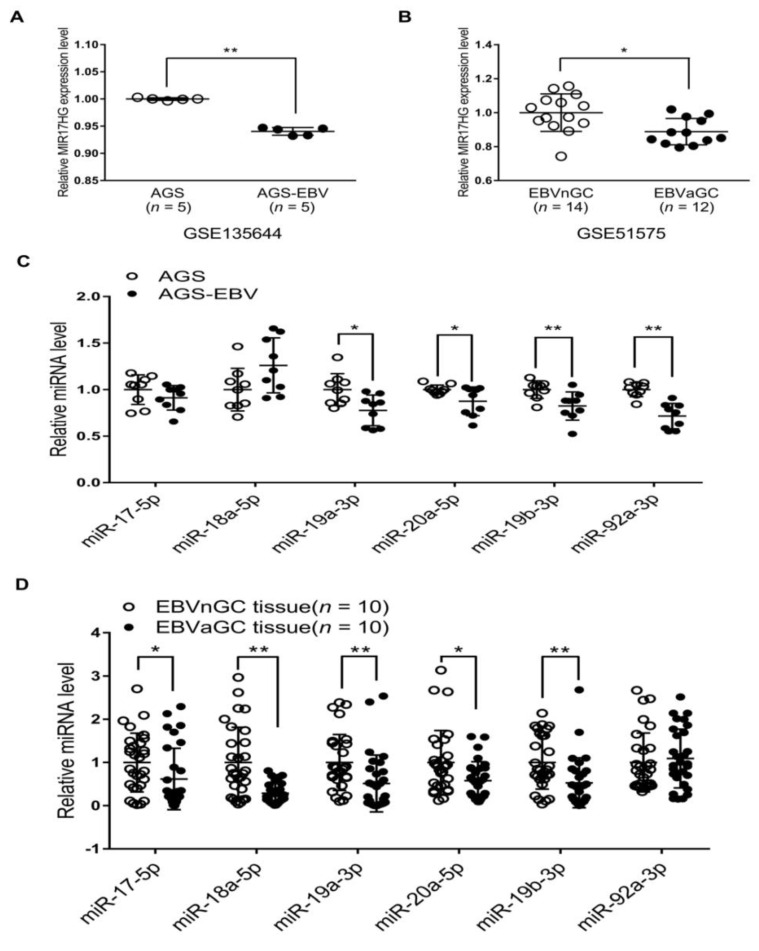
MIR17HG and miR-17-92 cluster miRNA expression in EBV-negative and EBV-positive samples. (**A**) Microarray analysis of MIR17HG expression level in AGS and AGS-EBV cells (*n* = 5). (**B**) Results of the GEO database analysis for MIR17HG expression in EBVnGC and EBVaGC cases. (**C**) and (**D**) The qRT-PCR analysis of miR-17-92 cluster miRNA expression was carried out using a SYBR Green qPCR kit. (**C**) Total RNA was extracted from AGS and AGS-EBV cells. Three sets of independent experiments were performed. (**D**) Total RNA was extracted from EBVaGC (*n* = 10) and EBVnGC (*n* = 10) samples, and qRT-PCR analyses were performed. The data were analyzed using Student’s *t*-test. Error bars indicate the SD. *, *p* < 0.05; **, *p* < 0.01.

**Figure 6 ijms-22-10936-f006:**
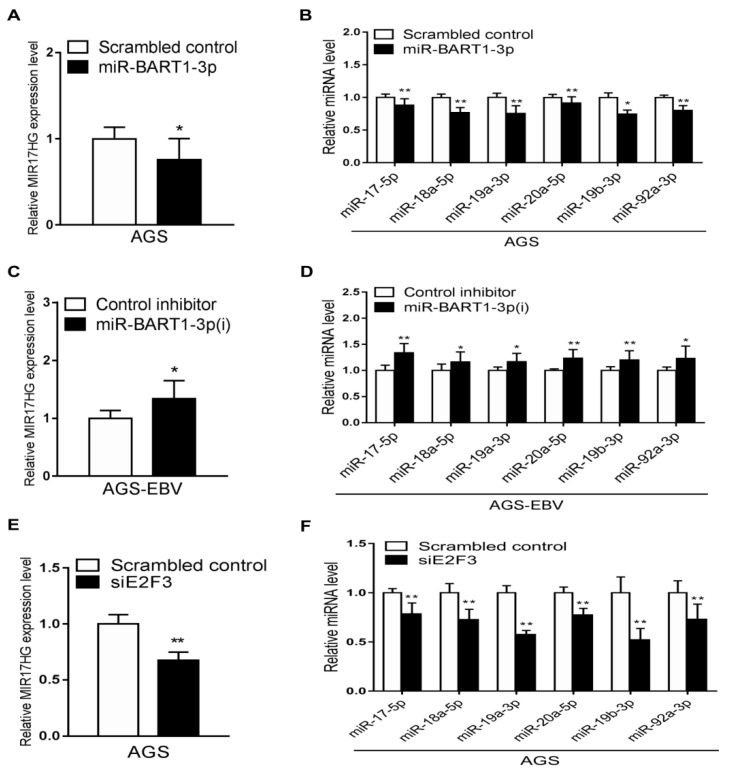
Effects of miR-BART1-3p and siE2F3 on miR-17-92 cluster miRNA expression. The AGS cells were transfected with 30 nM miR-BART1-3p mimic (**A**) and (**B**) or siE2F3 (**E**) and (**F**), while AGS-EBV cells were transfected with 30 nM LNA-miR-BART1-3p(i) (**C**) and (**D**). After 48 hr, the cells were harvested for RNA extraction. (**A**), (**C**), and (**E**) The expression of MIR17HG was measured by qRT-PCR. The MIR17HG expression was normalized to that of GAPDH and then expressed as a ratio compared with that of the control transfected cells. (**B**), (**D**), and (**F**) The qRT-PCR analysis of individual miRNAs encoded in the miR-17-92 cluster was carried out using a SYBR green qPCR kit. The expression of each miRNA was normalized to that of U6 and expressed as a ratio compared with that of the control transfected cells. The data were analyzed using Student’s *t*-test. Error bars indicate the SD (*n* = 3). *, *p* < 0.05; **, *p* < 0.01.

## Data Availability

The data presented in this study are available upon request from the corresponding author.

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
