# Peer review of "Epstein-Barr Virus miR-BART1-3p Regulates the miR-17-92 Cluster by Targeting E2F3"

_ijms, 2021, doi:10.3390/ijms222010936_

Round 1

Reviewer 1 Report

In their work, Park, Kim et al. described the role of EBV-encoded miR-BART1-3p in epithelial malignancies in a model of gastric cancer. In general, manuscript is well written and presents interesting results of clinical relevance.

The introduction is well written and presents the background of the study. However, I would suggest describe more deeply miRNAs, including EBV-encoded miRNAs, as well as their role of in cancer. Some recent reviews comprehensively describe these issues (PMID: 32038626, PMID: 32194570, PMID: 33321819, PMID: 29805308, PMID: 33921696).

The results section is clearly written and presented. Figures description is correct, however, the quality of the figures has to be improved.

Discussion should be more comprehensive and should include imporant papers regarding 1. the role of EBV-encoded miRNAs in oncogenesis 2. their role as prognostic factors (f.e. PMID: 31889055, PMID: 32582365).

However, several points need to be addressed before acceptance of this manuscript.

Major points:
1. In general, the quality of the figures is very poor and has to be improved.

2. Measure the level of miR-BART1-3p in AGS-EBV compared to AGS.

3. Measure the level of miR-BART1-3p after transfection with mimic-miR as well as the level of miR-BART1-3p after transfection with miR-inhibitor

4. Authors suggests that miR-BART1-3p acts via the E2F3-miR-17-92 cluster axis. However, they did not demonstrate that inhibition of miR-17-92 cluster has any effects in AGS cells. 

5. To confirm that the miR-BART1-3p - E2F3 - miR-17-92 axis has biological role, I expect the experiment with the AGS cell line transfected with both miR-BART1-3p mimic and miR-17-92 cluster miRNAs mimic. Lack of the differences in cell proliferation would confirm the existence of this axis. 

6. Is miR-BART1-3p expression correlated with OS or PFS of gastric cancer patients? Did authors analyze available datasets to answer this question?

7. I would suggest changing the graphs from bar graphs to box-and-whisker or dot-plot format to show data distribution.

Minor points:
1. Add the description of the cell lines (AGS and AGS-EBV) to the plots and graphs C-D and E-F.
2. Add numbers of biological replicates of 3. presented data to the description of figure 1.

3. Explain the EBVnGC abbreviation in the manuscript text.

4. L121 - protein level, not expression

5. L133 - the cell lines, not cell experiments

6. Add information about statistical tests used to calculate p value to the figure description.

7. Remove sentence from L86

Author Response

We appreciate kind advice from the reviewers with regard to the content of our paper. We carefully read all the comments and we are doing our best to revise the manuscript accordingly.

Minor points corrections are indicated in the manuscript in blue.

Reviewer 2 Report

The manuscript by Myung Chan Park and colleagues tells a story which is intriguing and with many potential implications, mostly regarding the putative interplay between miR-BART1-3p, E2F3, and miR-17-92 cluster miRNAs EBVaGC.
In the first part though, there are some important technical and conceptual issues that should be addressed before considering this manuscript for publication.
Please consider the following points section by section.

Introduction is well written and exhaustive; Materials and Methods are fine.

Sections: 2.1. miR-BART1-3p induces G0/G1 arrest and hinders the S-phase transition and 2.2. E2F3 expression is suppressed in EBVaGC and EBV-infected cells
My major concerns regard the use of MTT assay as unique tool for assessing cell proliferation and the use of Single parameter DNA analysis for determining cell cycle blocks.
There is strong debate if MTT/XTT assays are proper measure for cell proliferation, or if they are only a mere measure of cell viability. I understand that Authors report a progressive time-related increase of signal up to 72 hours post transfection, but this is not conclusive for me; the increase in measured OD could be to related to more efficient mitochondria activity for instance, and not (only) to an increase in number of cells. Have Authors also performed a proliferation profile? Have Authors applied correction factors which account for difference in the seeding of cells (at t=0) and for the AGS (theoretical) duplication time between replicates and between time points?
To be more convincing, in addition to the Metabolic assay (MTT), Authors have to provide also another type of cell proliferation assay, for example one based on DNA synthesis. I would exclude Ki67 assay because many tumor cells that express Ki67 undergo G1 arrest, exit the cell cycle or undergo apoptosis rather than complete the cell cycle.

“As transfection of miR-BART1-3p induced cell-cycle arrest at the G0/G1 phase in the AGS cells” maybe too strong? How do Authors know that cells transfected with miR-BART1-3p are effectively G0/G1 blocked? The differences between G0/G1 and S phase suggest a G0/G1 block, even if differences between phases are not tremendous though. Single parameter DNA analysis is not sufficient to discriminate between cells in very early or late S phase from cells in G1 and G2 phase. Authors should add a bromodeoxyuridine (BrdU) experiment or similar. Alternatively, Authors could show the downregulation of a couple of cell cycle progression markers such as CDK2, cyclin E1, CDK4, and cyclin D1 by western blot.

Section: 2.5. miR-17-92 cluster expression is suppressed in EBVaGC and EBV infected cells
This is a really interesting part, but Authors conclusion are weakened by the result proposed.
Standard deviation in Fig. 5D are extremely high. How many times have Authors performed the experiment? Please specify it in figure legend (as done for the other experiments). Have Authors tried to change the housekeeping gene? If the explanation of such variability is that reported in the Discussion section: “Therefore, the differences in miRNA expression observed in our experiments might have been related to the proteins and lncRNAs involved in post-transcriptional regulation of miRNA” Authors could anticipate it in the result section.
As for Figure 5C, I understand that the data are statistically significant, but I do not observe a significant reduction of individual miRNAs, or at least, not for all. I would suggest a reformulation.

2.6. miR-BART1-3p and siE2F3 suppress miR-17-92 cluster miRNA expression
To strengthen what the results shown in AGS cells transfected with the miR-BART1-3p and in AGS-EBV cells transfected with miR-BART1-3p(i), Authors could show some data on downstream target of miR-17-92 cluster. This would definitely increase the impact of this section.

Section: Discussion
This section is well written, but sub-sections seem to stand alone; Authors should make it more fluent. For example, is not clear to me, which type of latency there is in AGS-EBV infected cells (maybe I miss it). A few words on this could link better what Authors speculate about latency to what they have demonstrated with their data.

Minor points

- Fig. 1A and 3A correct the word “contorl” in the figure
- Page 3, line 120: Do Authors mean E2F2 and not E2F3?
- How did Authors manage to quantify the immunoblot signals of E2F3? Signals are very faint and blurry. Have Authors tried to change the conditions (Nitrocellulose instead of PVDF, anti-E2F3 from other companies)?
- Authors should move the description of MIR17HG  and miR-BART1-3pm from the figure legend to the main text to exemplify the comprehension.
- In section 4.3.” Transfection with miRNA mimic, miRNA inhibitor, and siRNA to E2F3” of Materials and Methods, Authors report thatFor the cell proliferation assay and FACS analyses, the cells were harvested 48 h after transfection.” Do they refer to MTT assay? If so there is contradiction with what is written in the MTT section that should be fixed.
- In the Discussion “The expression of the miR-17-92 cluster family was suppressed by miR-BART1-3p and by an siRNA specific to E2F3 but was enhanced by an inhibitor of miR-BART1-3p.” Again, I do not see suppression. This would be acceptable if more data will presented (see above).

Author Response

We appreciate kind advice from the reviewers with regard to the content of our paper. We carefully read all the comments and we are doing our best to revise the manuscript accordingly.

Minor comments are marked in red in the manuscript.

Reviewer 3 Report

In this manuscript, the authors described the role of miR-BART1-3p on cell cycle, proliferation, and miR-17-92 cluster expressions in AGS cells. They suggested that the effects of miR-BART1-3p are directly related to the downregulation of E2F3. The topic of this article is interesting, and standard methods appear to have been used capably. The results are presented and interpreted logically. However, there are several questions that the authors should solve.

Major Concerns

  1. The author used AGS and AGS-EBV cells to show the effect of miR-BART1-3p on cell proliferation and cell cycle arrest, and EBVnGC and EBVaGC are used for the analysis of E2F and miR-17-92 clusters. This reviewer wants to know the endogenous level changes of miR-BART1-3p in the line cells and patient samples. First, the authors have to show the difference in endogenous levels of miR-BART1-3p in the above cells. Second, the transfection efficiency of each oligo, including siRNAs, mutant, and inhibitors, used in this study needs to be shown. After that, it can be suggested that experimental conditions and the mechanisms suggested by the authors actually occur in EBVaGC.
  2. MicroRNAs such as miR-BART1-3p are known to regulate multiple genes. The author suggested that miR-BART1-3p regulates cell cycle and miR-17-93 cluster expressions through E2F3 suppression. However, the current data do not prove that miR-BART1-3p-mediated changes in cell proliferation and miR-17-92 cluster expressions are directly regulated via E2F3. To do so, the authors must show that E2F3 co-expression abrogates the effect of miR-BART1-3p in AGS as well as AGS-EBV (in the case of inhibitor).
  3. Figs. 3A -D, include miR-BART1-3p and inhibitor co-transfected group.
  4. Immunoblot quality should be improved. This reviewer wants to see all the uncropped images used in the statistical analysis with molecular markers.
  5. Illustration contents look good, but most of them are compressed too much. Make all in good shape.

Minor

  1. Describe what miR-BART1-3pm is in the methods or results.
  2. There are several mistypes and reference errors, for example, lines 72-74, 208, 281, 283, 291, 296, etc…

Author Response

We appreciate kind advice from the reviewers with regard to the content of our paper. We carefully read all the comments and we are doing our best to revise the manuscript accordingly.

Minor comments are marked in green in the manuscript.

Round 2

Reviewer 2 Report

I would like to thank the Authors for the corrections and modifications they made to the manuscript. 
The explanations they provided are sufficient and the experiments they are going to perform would definitely improve the manuscript. The time that Authors require to complete the experiments is acceptable.
Also typing errors and mistakes in the manuscript have been properly fixed.

I would like to clarify some of my questions as the Authors required.

⇒ We did not examine proliferation profiles related to the AGS cell line. As the reviewer suggested, we are planning to analyze cell proliferation by trypan blue exclusion assay. However, we need more time for the experiments (at least 2 weeks).

Since Authors answer is "no", they can ignore my second question (the application of correction factors), which was related to the Proliferation Profile (maybe I was not clear enough).
A Proliferation Profile performed with a mathematical model with proper correction factors would have spared Authors to perform further experiments; this would have strengthen the experiments already reported.
I suggest this review:

Angela M Jarrett, Ernesto A B F Lima, David A Hormuth 2nd, Matthew T McKenna, Xinzeng Feng, David A Ekrut, Anna Claudia M Resende, Amy Brock, Thomas E Yankeelov. Mathematical models of tumor cell proliferation: A review of the literature.  doi: 10.1080/14737140.2018.1527689. Epub 2018 Oct 22

or this article, even if is a bit old: 

J.L. Sherley, P. B. Stadler, J. Scott Stadler. A quantitative method for the analysis of mammalian cell proliferation in culture in terms of dividing and non-dividing cells.
https://doi.org/10.1111/j.1365-2184.1995.tb00062.x

Anyway the Trypan blue experiment is enough.

Considering the following sentence: “When the expression of the six individual miRNAs comprising the miR-17-92 cluster was assessed in the cell lines, four (miR-19a-3p, miR-20a-5p, miR-19b-3p, miR-92a-3p) were significantly reduced in AGS-EBV compared with AGS cells (Fig. 5C).”
The explanation of my request is that I do not see such significant (= impressive) reduction (even if data are statistically significant); my point is that this sentence is misleading.

Author Response

We appreciate kind advice from the reviewers with regard to the content of our paper. We carefully read all the comments and we are doing our best to revise the manuscript accordingly.

Reviewer 3 Report

The author needs an appropriate explanation for queries 2 and 3. If authors need time for the additional experiments, the authors may consider resubmitting the manuscripts after having enough time.

Rather than that, the reviewer suggests the following:

Since miRNA has multiple targets, there are additional targets of miR-BART1-3p other than E3F3. Because of this character of miRNAs, co-transfection of E2F3 may or may not abrogate the effect of miR-BART1-3p on the cell proliferation and miR-17-92 cluster expressions. For this reason, it may be difficult to explain whether the effects of miRNAs are ascribed to the direct suppression of specific targets. Nevertheless, interpretations that overcome these points can be found in many previous studies. Therefore, the authors must explain to the reviewer why the author's logic is appropriate, based on existing interpretations from previous studies, and why the E2F3 co-transfection experiment is insufficient in this study.

Inhibitor co-expression study suggested in query 3 was not considered by the authors.

Author Response

(The authors gave the same response as above.)

Round 3

Reviewer 2 Report

I would like Authors for the further corrections they made.

Reviewer 3 Report

The Authors responded appropriately to the reviewer's criticisms. Coexpression experiments are not necessary at this moment.

The revised manuscript is now suitable for publication in its present form.